# Uncertainty-guided Continual Learning with Bayesian Neural Networks

**Sayna Ebrahimi**[*]    **Mohamed Elhoseiny**[†]    **Trevor Darrell**    **Marcus Rohrbach**
UC Berkeley           KAUST, Stanford University   UC Berkeley       Facebook AI Research

## Abstract

Continual learning aims to learn new tasks without forgetting previously learned ones. This is especially challenging when one cannot access data from previous tasks and when the model has a fixed capacity. Current regularization-based continual learning algorithms need an external representation and extra computation to measure the parameters' *importance*. In contrast, we propose Uncertainty-guided Continual Bayesian Neural Networks (UCB), where the learning rate adapts according to the uncertainty defined in the probability distribution of the weights in networks. Uncertainty is a natural way to identify *what to remember* and *what to change* as we continually learn, and thus mitigate catastrophic forgetting. We also show a variant of our model, which uses uncertainty for weight pruning and retains task performance after pruning by saving binary masks per tasks. We evaluate our UCB approach extensively on diverse object classification datasets with short and long sequences of tasks and report superior or on-par performance compared to existing approaches. Additionally, we show that our model does not necessarily need task information at test time, i.e. it does not presume knowledge of which task a sample belongs to.

## 1 Introduction

Humans can easily accumulate and maintain knowledge gained from previously observed tasks, and continuously learn to solve new problems or tasks. Artificial learning systems typically forget prior tasks when they cannot access all training data at once but are presented with task data in sequence.

Overcoming these challenges is the focus of *continual learning*, sometimes also referred to as *lifelong learning* or *sequential learning*. *Catastrophic forgetting* (McCloskey & Cohen, 1989; McClelland et al., 1995) refers to the significant drop in the performance of a learner when switching from a trained task to a new one. This phenomenon occurs because trained parameters on the initial task change in favor of learning new objectives.

Given a network of limited capacity, one way to address this problem is to identify the importance of each parameter and penalize further changes to those parameters that were deemed to be important for the previous tasks (Kirkpatrick et al., 2017; Aljundi et al., 2018; Zenke et al., 2017). An alternative is to freeze the most important parameters and allow future tasks to only adapt the remaining parameters to new tasks (Mallya & Lazebnik, 2018). Such models rely on the explicit parametrization of importance. We propose here implicit uncertainty-guided importance representation.

Bayesian approaches to neural networks (MacKay, 1992b) can potentially avoid some of the pitfalls of explicit parameterization of importance in regular neural networks. Bayesian techniques, naturally account for uncertainty in parameters estimates. These networks represent each parameter with a distribution defined by a mean and variance over possible values drawn from a shared latent probability distribution (Blundell et al., 2015). Variational inference can approximate posterior distributions using Monte Carlo sampling for gradient estimation. These networks act like ensemble methods in that they reduce the prediction variance but only use twice the number of parameters present in a regular neural network. We propose to use the predicted mean and variance of the latent distributions to characterize the importance of each parameter. We perform continual learning with

---

[*]Corresponding author: `sayna@berkeley.edu`
[†]Work done while at Facebook AI Research

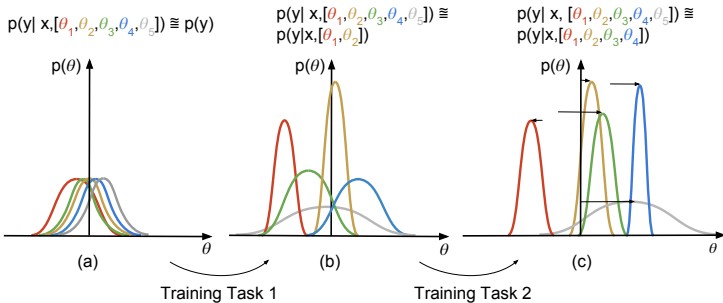

Figure 1: Illustration of the evolution of weight distributions – uncertain weights adapt more quickly – when learning two tasks using UCB. (a) weight parameter initialized by distributions initialized with mean and variance values randomly sampled from $\mathcal{N}(0, 0.1)$. (b) posterior distribution after learning task one; while $\theta_1$ and $\theta_2$ exhibit lower uncertainties after learning the first task, $\theta_3$, $\theta_4$, and $\theta_5$ have larger uncertainties, making them available to learn more tasks. (c) a second task is learned using higher learning rates for previously uncertain parameters ($\theta_1$, $\theta_2$, $\theta_3$, and $\theta_4$) while learning rates for $\theta_1$ and $\theta_2$ are reduced. Size of the arrows indicate the magnitude of the change of the distribution mean upon gradient update.

Bayesian neural networks by controlling the learning rate of each parameter as a function of its uncertainty. Figure 1 illustrates how posterior distributions evolve for certain and uncertain weight distributions while learning two consecutive tasks. Intuitively, the more uncertain a parameter is, the more learnable it can be and therefore, larger gradient steps can be taken for it to learn the current task. As a hard version of this regularization technique, we also show that pruning, i.e., preventing the most important model parameters from any change and learning new tasks with the remaining parameters, can be also integrated into UCB. We refer to this method as UCB-P.

**Contributions:** We propose to perform continual learning with Bayesian neural networks and develop a new method which exploits the inherent measure of uncertainty therein to adapt the learning rate of individual parameters (Sec. 4). Second, we introduce a hard-threshold variant of our method that decides which parameters to freeze (Sec. 4.2). Third, in Sec. 5, we extensively validate our approach experimentally, comparing it to prior art both on single datasets split into different tasks, as well as for the more difficult scenario of learning a sequence of different datasets. Forth, in contrast to most prior work, our approach does not rely on knowledge about task boundaries at inference time, which humans do not need and might not be always available. We show in Sec. 6 that our approach naturally supports this scenario and does not require task information at test time, sometimes also referred to as a "single head" scenario for all tasks. We refer to evaluation metric of a "single head" model without task information at test time as "generalized accuracy". Our code is available at https://github.com/SaynaEbrahimi/UCB.

## 2 RELATED WORK

Conceptually, approaches to continual learning can be divided into the following categories: dynamic architectural methods, memory-based methods, and regularization methods.

**Dynamic architectural methods**: In this setting, the architecture grows while keeping past knowledge fixed and storing new knowledge in different forms such as additional layers, nodes, or modules. In this approach, the objective function remains fixed whereas the model capacity grows –often exponentially– with the number of tasks. Progressive networks (Rusu et al., 2016; Schwarz et al., 2018) was one of the earliest works in this direction and was successfully applied to reinforcement learning problems; the base architecture was duplicated and lateral connections added in response to new tasks. Dynamically Expandable Network (DEN) (Yoon et al., 2018) also expands its network by selecting *drifting* units and retraining them on new tasks. In contrast to our method, these approaches require the architecture grow with each new task.

**Memory-based methods:** In this regime, previous information is partially stored to be used later as a form of *rehearsal* (Robins, 1995). Gradient episodic memory (GEM) (Lopez-Paz et al., 2017) uses this idea to store the data at the end of each episode to be used later to prevent gradient updates from deviating from their previous values. GEM also allows for positive backward knowledge transfer, i.e,

an improvement on previously learned tasks, and it was the first method capable of learning using a single training example. Recent approaches in this category have mitigated forgetting by using external data combined with distillation loss and/or confidence-based sampling strategies to select the most representative samples. (Castro et al., 2018; Wu et al., 2019; Lee et al., 2019)

**Regularization methods**: In these approaches, significant changes to the representation learned for previous tasks are prevented. This can be performed through regularizing the objective function or directly enforced on weight parameters. Typically, this *importance* measure is engineered to represent the importance of each parameter. Inspired by Bayesian learning, in elastic weight consolidation (EWC) method (Kirkpatrick et al., 2017) important parameters are those to have the highest in terms of the Fisher information matrix. In Synaptic Intelligence (SI) (Zenke et al., 2017) this parameter importance notion is engineered to correlate with the loss function: parameters that contribute more to the loss are more important. Similar to SI, Memory-aware Synapses (MAS) (Aljundi et al., 2018) proposed an online way of computing importance adaptive to the test set using the change in the model outputs w.r.t the inputs. While all the above algorithms are task-dependent, in parallel development to this work, (Aljundi et al., 2019) has recently investigated task-free continual learning by building upon MAS and using a protocol to update the weights instead of waiting until the tasks are finished. PackNet (Mallya & Lazebnik, 2018) used iterative pruning to fully restrict gradient updates on important weights via binary masks. This method requires knowing which task is being tested to use the appropriate mask. PackNet also ranks the weight importance by their magnitude which is not guaranteed to be a proper importance indicative. HAT (Serra et al., 2018) identifies important neurons by learning an attention vector to the task embedding to control the gradient propagation. It maintains the information learned on previous tasks using an almost-binary mask per previous tasks.

**Bayesian approaches:** Using Bayesian approach in learning neural networks has been studied for few decades (MacKay, 1992b;a). Several approaches have been proposed for Bayesian neural networks, based on, e.g., the Laplace approximation (MacKay, 1992a), Hamiltonian Monte Carlo (Neal, 2012), variational inference (Hinton & Van Camp, 1993; Graves, 2011), and probabilistic backpropagation (Hernández-Lobato & Adams, 2015). Variational continual learning (Nguyen et al., 2018) uses Bayesian inference to perform continual learning where new posterior distribution is simply obtained by multiplying the previous posterior by the likelihood of the dataset belonging to the new task. They also showed that by using a core-set, a small representative set of data from previous tasks, VCL can experience less forgetting. In contrast, we rely on Bayesian neural networks to use their predictive uncertainty to perform continual learning. Moreover, we do not use episodic memory or any other way to access or store previous data in our approach.

**Natural gradient descent methods:** A fast natural gradient descent method for variational inference was introduced in (Khan & Nielsen, 2018) in which, the Fisher Information matrix is approximated using the generalized Gauss-Newton method. In contrast, in our work, we use classic gradient descent. Although second order optimization algorithms are proven to be more accurate than the first order methods, they add considerable computational cost. Tseran et al. (2018); Chen et al. (2019) both investigate the effect of natural gradient descent methods as an alternative to classic gradient descent used in VCL and EWC methods. GNG (Chen et al., 2019) uses Gaussian natural gradients in the Adam optimizer (Kingma & Ba, 2014) in the framework of VCL because as opposed to conventional gradient methods which perform in Euclidian space, natural gradients cause a small difference in terms of distributions following the changes in parameters in the Riemannian space. Similar to VCL, they obtained their best performance by adding a coreset of previous examples. Tseran et al. (2018) introduce two modifications to VCL called Natural-VCL (N-VCL) and VCL-Vadam. N-VCL (Tseran et al., 2018) uses a Gauss-Newton approximation introduced by (Schraudolph, 2002; Graves, 2011) to estimate the VCL objective function and used natural gradient method proposed in (Khan et al., 2018) to exploit the Riemannian geometry of the variational posterior by scaling the gradient with an adaptive learning rate equal to $\sigma^{-2}$ obtained by approximating the Fisher Information matrix in an online fashion. VCL-Vadam (Tseran et al., 2018) is a simpler version of N-VCL to trade-off accuracy for simplicity which uses Vadam (Khan et al., 2018) to update the gradients by perturbing the weights with a Gaussian noise using a reparameterization trick and scaling by $\sigma^{-1}$ instead of its squared. N-VCL/VCL-Vadam both use variational inference to adapt the learning rate within Adam optimizer at every time step, whereas in our method below, gradient decent is used with constant learning rate during each task where learning rate scales with uncertainty only after finishing a task. We show extensive comparison with state-of-the-art results on short and relatively long sequence of vision datasets with Bayesian *convolutional* neural networks, whereas VCL-Vadam only rely on

multi-layer perceptron networks. We also like to highlight that this is the first work which evaluates and shows the working of convolutional Bayesian Neural Networks rather than only fully connected MLP models for continual learning.

# 3 BACKGROUND: VARIATIONAL BAYES-BY-BACKPROP

In this section, we review the Bayes-by-Backprop (BBB) framework which was introduced by (Blundell et al., 2015); to learn a probability distribution over network parameters. (Blundell et al., 2015) showed a back-propagation-compatible algorithm which acts as a regularizer and yields comparable performance to dropout on the MNIST dataset. In Bayesian models, latent variables are drawn from a prior density $p(\mathbf{w})$ which are related to the observations through the likelihood $p(\mathbf{x}|\mathbf{w})$. During inference, the posterior distribution $p(\mathbf{w}|\mathbf{x})$ is computed conditioned on the given input data. However, in practice, this probability distribution is intractable and is often estimated through approximate inference. Markov Chain Monte Carlo (MCMC) sampling (Hastings, 1970) has been widely used and explored for this purpose, see (Robert & Casella, 2013) for different methods under this category. However, MCMC algorithms, despite providing guarantees for finding asymptotically exact samples from the target distribution, are not suitable for large datasets and/or large models as they are bounded by speed and scalability issues. Alternatively, variational inference provides a faster solution to the same problem in which the posterior is approximated using optimization rather than being sampled from a chain (Hinton & Van Camp, 1993).Variational inference methods always take advantage of fast optimization techniques such as stochastic methods or distributed methods, which allow them to explore data models quickly. See (Blei et al., 2017) for a complete review of the theory and (Shridhar et al., 2018) for more discussion on how to use Bayes by Backprop (BBB) in convolutioal neural networks.

## 3.1 BAYES BY BACKPROP (BBB)

Let $\mathbf{x} \in \mathbb{R}^n$ be a set of observed variables and $\mathbf{w}$ be a set of latent variables. A neural network, as a probabilistic model $P(\mathbf{y}|\mathbf{x}, \mathbf{w})$, given a set of training examples $\mathcal{D} = (\mathbf{x}, \mathbf{y})$ can output $\mathbf{y}$ which belongs to a set of classes by using the set of weight parameters $\mathbf{w}$. Variational inference aims to calculate this conditional probability distribution over the latent variables by finding the closest proxy to the exact posterior by solving an optimization problem.

We first assume a family of probability densities over the latent variables $\mathbf{w}$ parametrized by $\theta$, i.e., $q(\mathbf{w}|\theta)$. We then find the closest member of this family to the true conditional probability of interest $P(\mathbf{w}|\mathcal{D})$ by minimizing the Kullback-Leibler (KL) divergence between $q$ and $P$ which is equivalent to minimizing variational free energy or maximizing the expected lower bound:

$$\theta^* = \arg\min_\theta \text{KL}\big(q(\mathbf{w}|\theta)\|P(\mathbf{w}|\mathcal{D})\big) \tag{1}$$

The objective function can be written as:

$$\mathcal{L}_{BBB}(\theta, \mathcal{D}) = \text{KL}\big[q(\mathbf{w}|\theta)\|P(\mathbf{w})\big] - \mathbb{E}_{q(\mathbf{w}|\theta)}\big[\log(P(\mathcal{D}|\mathbf{w}))\big] \tag{2}$$

Eq. 2 can be approximated using $N$ Monte Carlo samples $\mathbf{w}_i$ from the variational posterior (Blundell et al., 2015):

$$\mathcal{L}_{BBB}(\theta, \mathcal{D}) \approx \sum_{i=1}^{N} \log q(\mathbf{w}_i|\theta) - \log P(\mathbf{w}_i) - \log(P(\mathcal{D}|\mathbf{w}_i)) \tag{3}$$

We assume $q(\mathbf{w}|\theta)$ to have a Gaussian pdf with diagonal covariance and parametrized by $\theta = (\mu, \rho)$. A sample weight of the variational posterior can be obtained by sampling from a unit Gaussian and reparametrized by $\mathbf{w} = \mu + \sigma \circ \epsilon$ where $\epsilon$ is the noise drawn from unit Gaussian, and $\circ$ is a pointwise multipliation. Standard deviation is parametrized as $\sigma = \log(1 + \exp(\rho))$ and thus is always positive. For the prior, as suggested by Blundell et al. (2015), a scale mixture of two Gaussian pdfs are chosen which are zero-centered while having different variances of $\sigma_1^2$ and $\sigma_2^2$. The uncertainty obtained for every parameter has been successfully used in model compression (Han et al., 2015) and uncertainty-based exploration in reinforcement learning (Blundell et al., 2015). In this work we propose to use this framework to learn sequential tasks without forgetting using per-weight uncertainties.

# 4 UNCERTAINTY-GUIDED CONTINUAL LEARNING IN BAYESIAN NEURAL NETWORKS

In this section, we introduce Uncertainty-guided Continual learning approach with Bayesian neural networks (UCB), which exploits the estimated uncertainty of the parameters' posterior distribution to regulate the change in "important" parameters both in a soft way (Section 4.1) or setting a hard threshold (Section 4.2).

## 4.1 UCB WITH LEARNING RATE REGULARIZATION

A common strategy to perform continual learning is to reduce forgetting by regularizing further changes in the model representation based on parameters' *importance*. In UCB the regularization is performed with the learning rate such that the learning rate of each parameter and hence its gradient update becomes a function of its *importance*. As shown in the following equations, in particular, we scale the learning rate of $\mu$ and $\rho$ for each parameter distribution inversely proportional to its importance $\Omega$ to reduce changes in important parameters while allowing less important parameters to alter more in favor of learning new tasks.

$$\alpha_\mu \leftarrow \alpha_\mu / \Omega_\mu \tag{4}$$
$$\alpha_\rho \leftarrow \alpha_\rho / \Omega_\rho \tag{5}$$

The core idea of this work is to base the definition of importance on the well-defined uncertainty in parameters distribution of Bayesian neural networks, i.e., setting the *importance* to be inversely proportional to the standard deviation $\sigma$ which represents the parameter uncertainty in the Baysian neural network:

$$\Omega \propto 1/\sigma \tag{6}$$

We explore different options to set $\Omega$ in our ablation study presented in Section A.2 of the appendix, Table 1. We empirically found that $\Omega_\mu = 1/\sigma$ and not adapting the learning rate for $\rho$ (i.e. $\Omega_\rho = 1$) yields the highest accuracy and the least forgetting.

The key benefit of UCB with learning rate as the regularizer is that it neither requires additional memory, as opposed to pruning technique nor tracking the change in parameters with respect to the previously learned task, as needed in common weight regularization methods.

More importantly, this method does not need to be aware of task switching as it only needs to adjust the learning rates of the means in the posterior distribution based on their current uncertainty. The complete algorithm for UCB is shown in Algorithm 1 with parameter update function given in Algorithm 2.

## 4.2 UCB USING WEIGHT PRUNING (UCB-P)

In this section, we introduce a variant of our method, UCB-P, which is related to recent efforts in weight pruning in the context of reducing inference computation and network compression (Liu et al., 2017; Molchanov et al., 2016). More specifically, weight pruning has been recently used in continual learning (Mallya & Lazebnik, 2018), where the goal is to continue learning multiple tasks using a single network's capacity. (Mallya & Lazebnik, 2018) accomplished this by freeing up parameters deemed to be unimportant to the current task according to their magnitude. Forgetting is prevented in pruning by saving a task-specific binary mask of important vs. unimportant parameters. Here, we adapt pruning to Bayesian neural networks. Specifically, we propose a different criterion for measuring importance: the statistically-grounded uncertainty defined in Bayesian neural networks.

Unlike regular deep neural networks, in a BBB model weight parameters are represented by probability distributions parametrized by their mean and standard deviation. Similar to (Blundell et al., 2015), in order to take into account both mean and standard deviation, we use the signal-to-noise ratio (SNR) for each parameter defined as

$$\Omega = \text{SNR} = |\mu|/\sigma \tag{7}$$

---

**Algorithm 1** Uncertainty-guided Continual Learning with Bayesian Neural Networks UCB

---

1: **Require** Training data for all tasks $\mathcal{D} = (\mathbf{x}, \mathbf{y})$, $\mu$ (mean of posterior), $\rho$, $\sigma_1$ and $\sigma_2$ (std for the scaled mixture Gaussian pdf of prior), $\pi$ (weighting factor for prior), $N$ (number of samples in a mini-batch), $M$ (Number of minibatches per epoch), initial learning rate ($\alpha_0$)
2:    $\alpha_\mu = \alpha_\rho = \alpha_0$
3: **for** every task **do**
4:    **repeat**
5:      $\epsilon \sim \mathcal{N}(0, I)$
6:      $\sigma = \log(1 + \exp(\rho))$                           $\triangleright$ Ensures $\sigma$ is always positive
7:      $\mathbf{w} = \mu + \sigma \circ \epsilon$            $\triangleright \mathbf{w} = \{\mathbf{w}_1, \ldots, \mathbf{w}_i, \ldots, \mathbf{w}_N\}$ posterior samples of weights
8:      $l_1 = \sum_{i=1}^{N} \log \mathcal{N}(\mathbf{w}_i | \mu, \sigma^2)$              $\triangleright l_1 :=$ Log-posterior
9:      $l_2 = \sum_{i=1}^{N} \log \left( \pi \mathcal{N}(\mathbf{w}_i \mid 0, \sigma_1^2) + (1 - \pi)\mathcal{N}(\mathbf{w}_i \mid 0, \sigma_2^2) \right)$    $\triangleright l_2 :=$ Log-prior
10:     $l_3 = \sum_{i=1}^{N} \log(p(\mathcal{D}|\mathbf{w}_i))$              $\triangleright l_3 :=$ Log-likelihood of data
11:     $\mathcal{L}_{BBB} = \frac{1}{M}(l_1 - l_2 - l_3)$
12:     $\mu \leftarrow \mu - \alpha_\mu \nabla \mathcal{L}_{BBB_\mu}$
13:     $\rho \leftarrow \rho - \alpha_\rho \nabla \mathcal{L}_{BBB_\rho}$
14:    **until** loss plateaus
15:    $\alpha_\mu, \alpha_\rho \leftarrow$ LearningRateUpdate$(\alpha_\mu, \alpha_\rho, \sigma, \mu)$      $\triangleright$ See Algorithm 2 for UCB and 3 for UCB-P
16: **end for**

---

| **Algorithm 2** LearningRateUpdate in UCB | **Algorithm 3** LearningRateUpdate in UCB-P |
|---|---|
| 1: **function** LearningRateUpdate$(\alpha_\mu, \alpha_\rho, \sigma)$ | 1: **function** LearningRateUpdate$(\alpha_\mu, \alpha_\rho, \sigma, \mu)$ |
| 2:    **for** each parameter **do** | 2:    **for** each parameter $j$ in each layer $l$ **do** |
| 3:      $\Omega_\mu \leftarrow 1/\sigma$ | 3:      $\Omega \leftarrow |\mu|/\sigma$     $\triangleright$ Signal to noise ratio |
| 4:      $\Omega_\rho \leftarrow 1$ | 4:      **if** $\Omega[j] \in$ top $p\%$ of $\Omega$s in $l$ **then** |
| 5:      $\alpha_\mu \leftarrow \alpha_\mu / \Omega_\mu$ | 5:        $\alpha_\mu = \alpha_\rho = 0$ |
| 6:      $\alpha_\rho \leftarrow \alpha_\rho / \Omega_\rho$ | 6:      **end if** |
| 7:    **end for** | 7:    **end for** |
| 8: **end function** | 8: **end function** |

SNR is a commonly used measure in signal processing to distinguish between "useful" information from unwanted noise contained in a signal. In the context of neural models, the SNR can be thought as an indicative of parameter importance; the higher the SNR, the more effective or important the parameter is to the model predictions for a given task.

UCB-P, as shown in Algorithms 1 and 3, is performed as follows: for every layer, convolutional or fully-connected, the parameters are ordered by their SNR value and those with the lowest importance are pruned (set to zero). The pruned parameters are marked using a binary mask so that they can be used later in learning new tasks whereas the important parameters remain fixed throughout training on future tasks. Once a task is learned, an associated binary mask is saved which will be used during inference to recover key parameters and hence the exact performance to the desired task.

The overhead memory per parameter in encoding the mask as well as saving it on the disk is as follows. Assuming we have $n$ tasks to learn using a single network, the total number of required bits to encode an accumulated mask for a parameter is at max $\log_2 n$ bits assuming a parameter deemed to be important from task 1 and kept being encoded in the mask.

## 5 RESULTS

### 5.1 EXPERIMENTAL SETUP

**Datasets:** We evaluate our approach in two common scenarios for continual learning: 1) class-incremental learning of a single or two randomly alternating datasets, where each task covers only a subset of the classes in a dataset, and 2) continual learning of multiple datasets, where each task is a dataset. We use Split MNIST with 5 tasks (5-Split MNIST) similar to (Nguyen et al., 2018; Chen et al., 2019; Tseran et al., 2018) and permuted MNIST (Srivastava et al., 2013) for class incremental learning with similar experimental settings as used in (Serra et al., 2018; Tseran et al., 2018). Furthermore, to have a better understanding of our method, we evaluate our approach on continually learning a sequence of 8 datasets with different distributions using the identical sequence

as in (Serra et al., 2018), which includes FaceScrub (Ng & Winkler, 2014), MNIST, CIFAR100, NotMNIST (Bulatov, 2011), SVHN (Netzer et al., 2011), CIFAR10, TrafficSigns (Stallkamp et al., 2011), and FashionMNIST (Xiao et al., 2017). Details of each are summarized in Table 4 in appendix. No data augmentation of any kind has been used in our analysis.

**Baselines:** Within the Bayesian framework, we compare to three models which do not incorporate the importance of parameters, namely fine-tuning, feature extraction, and joint training. In fine-tuning (BBB-FT), training continues upon arrival of new tasks without any forgetting avoidance strategy. Feature extraction, denoted as (BBB-FE), refers to freezing all layers in the network after training the first task and training only the last layer for the remaining tasks. In joint training (BBB-JT) we learn all the tasks jointly in a multitask learning fashion which serves as the upper bound for average accuracy on all tasks, as it does not adhere to the continual learning scenario. We also perform the counterparts for FT, FE, and JT using ordinary neural networks and denote them as ORD-FT, ORD-FE, and ORD-JT. From the prior work, we compare with state-of-the-art approaches including Elastic Weight Consolidation (EWC) (Kirkpatrick et al., 2017), Incremental Moment Matching (IMM) (Lee et al., 2017), Learning Without Forgetting (LWF) (Li & Hoiem, 2016), Less-Forgetting Learning (LFL) (Jung et al., 2016), PathNet (Fernando et al., 2017), Progressive neural networks (PNNs) (Rusu et al., 2016), and Hard Attention Mask (HAT) (Serra et al., 2018) using implementations provided by (Serra et al., 2018). On Permuted MNIST results for SI (Zenke et al., 2017) are reported from (Serra et al., 2018). On Split and Permuted MNIST, results for VCL (Nguyen et al., 2018) are obtained using their original provided code whereas for VCL-GNG (Chen et al., 2019) and VCL-Vadam (Tseran et al., 2018) results are reported from the original work without re-implementation. Because our method lies into the regularization-based regime, we only compare against baselines which do not benefit from episodic or coreset memory.

**Hyperparameter tuning:** Unlike commonly used tuning techniques which use a validation set composed of *all* classes in the dataset, we only rely on the first two task and their validations set, similar to the setup in (Chaudhry et al., 2019). In all our experiments we consider a $0.15$ split for the validation set on the first two tasks. After tuning, training starts from the beginning of the sequence. Our scheme is different from (Chaudhry et al., 2019), where the models are trained on the first (e.g. three) tasks for validation and then training is restarted for the remaining ones and the reported performance is only on the remaining tasks.

**Training details:** It is important to note that in all our experiments, *no pre-trained model is used*. We used stochastic gradient descent with a batch size of $64$ and a learning rate of $0.01$, decaying it by a factor of $0.3$ once the loss plateaued. Dataset splits and batch shuffle are identically in all UCB experiments and all baselines.

**Pruning procedure and mask size**: Once a task is learned, we compute the performance drop for a set of arbitrary pruning percentages from the maximum training accuracy achieved when no pruning is applied. The pruning portion is then chosen using a threshold beyond which the performance drop is not accepted. Mask size is chosen without having the knowledge of how many tasks to learn in the future. Upon learning each task we used a uniform distribution of pruning ratios (50-100%) and picked the ratio resulted in at most $1\%$, $2\%$, and $3\%$ forgetting for MNIST, CIFAR, and 8tasks experiments, respectively. We did not tune this parameter because in our hyperparameter tuning, we only assume we have validation sets of the first two tasks.

**Parameter regularization and importance measurement:** Table 1 ablates different ways to compute the *importance* $\Omega$ of an parameter in Eq. 4 and 5. As shown in Table 1 the configuration that yields the highest accuracy and the least forgetting (maximum BWT) occurs when the learning rate regularization is performed only on $\mu$ of the posteriors using $\Omega_\mu = 1/\sigma$ as the importance and $\Omega_\rho = 1$.

**Performance measurement:** Let $n$ be the total number of tasks. Once all are learned, we evaluate our model on all $n$ tasks. ACC is the average test classification accuracy across all tasks. To measure forgetting we report backward transfer, BWT, which indicates how much learning new tasks has influenced the performance on previous tasks. While $\text{BWT} < 0$ directly reports *catastrophic forgetting*, $\text{BWT} > 0$ indicates that learning new tasks has helped with the preceding tasks. Formally, BWT and ACC are as follows:

$$\text{BWT} = \frac{1}{n}\sum_{i=1}^{n} R_{i,n} - R_{i,i}, \quad \text{ACC} = \frac{1}{n}\sum_{i=1}^{n} R_{i,n} \tag{8}$$

Table 1: Variants of learning rate regularization and importance measurement on 2-Split MNIST

| Method | $\mu$ | $\rho$ | Importance $\Omega$ | BWT (%) | ACC (%) |
|---|---|---|---|---|---|
| UCB | x | - | $1/\sigma$ | 0.00 | 99.2 |
| UCB | - | x | $1/\sigma$ | $-0.04$ | 98.7 |
| UCB | x | x | $1/\sigma$ | $-0.02$ | 98.0 |
| UCB | x | - | $|\mu|/\sigma$ | $-0.03$ | 98.4 |
| UCB | - | x | $|\mu|/\sigma$ | $-0.52$ | 98.7 |
| UCB | x | x | $|\mu|/\sigma$ | $-0.32$ | 98.8 |
| UCB-P | x | x | $|\mu|/\sigma$ | $-0.01$ | 99.0 |
| UCB-P | x | x | $1/\sigma$ | $-0.01$ | 98.9 |

Table 2: Continually learning on different datasets. BWT and ACC in %. (*) denotes that methods do not adhere to the continual learning setup: BBB-JT and ORD-JT serve as the upper bound for ACC for BBB/ORD networks, respectively. ‡ denotes results reported by (Serra et al., 2018). † denotes the result reported from original work. BWT was not reported in ‡ and †. All others results are (re)produced by us and are averaged over 3 runs with standard deviations given in Section A.3 of the appendix.

(a) 5-Split MNIST, 5 tasks.

| Method | BWT | ACC |
|---|---|---|
| VCL-Vadam† | - | 99.17 |
| VCL-GNG† | - | 96.50 |
| VCL | -0.56 | 98.20 |
| IMM | -11.20 | 88.54 |
| EWC | -4.20 | 95.78 |
| HAT | 0.00 | 99.59 |
| ORD-FT | -9.18 | 90.60 |
| ORD-FE | 0.00 | 98.54 |
| BBB-FT | -6.45 | 93.42 |
| BBB-FE | 0.00 | 98.76 |
| UCB-P (Ours) | -0.72 | 99.32 |
| **UCB (Ours)** | 0.00 | **99.63** |
| ORD-JT* | 0.00 | 99.78 |
| BBB-JT* | 0.00 | 99.87 |

(b) Permuted MNIST, 10 permutations.

| Method | #Params | BWT | ACC |
|---|---|---|---|
| SI ‡ | 0.1M | - | 86.0 |
| EWC ‡ | 0.1M | - | 88.2 |
| HAT ‡ | 0.1M | - | 91.6 |
| VCL-Vadam† | 0.1M | - | 86.34 |
| VCL-GNG† | 0.1M | - | 90.50 |
| VCL | 0.1M | -7.90 | 88.80 |
| UCB (Ours) | 0.1M | -0.38 | 91.44 |
| LWF | 1.9M | -31.17 | 65.65 |
| IMM | 1.9M | -7.14 | 90.51 |
| HAT | 1.9M | 0.03 | 97.34 |
| BBB-FT | 1.9M | -0.58 | 90.01 |
| BBB-FE | 1.9M | 0.02 | 93.54 |
| UCB-P (Ours) | 1.9M | -0.95 | 97.24 |
| **UCB (Ours)** | 1.9M | 0.03 | **97.42** |
| BBB-JT* | 1.9M | 0.00 | 98.12 |

(c) Alternating CIFAR10/100

| Method | BWT | ACC |
|---|---|---|
| PathNet | 0.00 | 28.94 |
| LWF | -37.9 | 42.93 |
| LFL | -24.22 | 47.67 |
| IMM | -12.23 | 69.37 |
| PNN | 0.00 | 70.73 |
| EWC | -1.53 | 72.46 |
| HAT | -0.04 | 78.32 |
| BBB-FE | -0.04 | 51.04 |
| BBB-FT | -7.43 | 68.89 |
| UCB-P (Ours) | -1.89 | 77.32 |
| **UCB (Ours)** | -0.72 | **79.44** |
| BBB-JT* | 1.52 | 83.93 |

(d) Sequence of 8 tasks

| Method | BWT | ACC |
|---|---|---|
| LFL | -10.0 | 8.61 |
| PathNet | 0.00 | 20.22 |
| LWF | -54.3 | 28.22 |
| IMM | -38.5 | 43.93 |
| EWC | -18.04 | 50.68 |
| PNN | 0.00 | 76.78 |
| HAT | -0.14 | 81.59 |
| BBB-FT | -23.1 | 43.09 |
| BBB-FE | -0.01 | 58.07 |
| UCB-P (Ours) | -2.54 | 80.38 |
| **UCB (Ours)** | -0.84 | **84.04** |
| BBB-JT* | -1.2 | 84.1 |

where $R_{i,n}$ is the test classification accuracy on task $i$ after sequentially finishing learning the $n^{\text{th}}$ task. Note that in UCB-P, $R_{i,i}$ refers the test accuracy on task $i$ before pruning and $R_{i,n}$ after pruning which is equivalent to the end of sequence performance. In Section 6, we show that our UCB model can be used when tasks labels are not available at inference time by training it with a "single head" architecture with a sum of number of classes for all tasks. We refer to the ACC measured for this scenario as "Generalized Accuracy".

## 5.2 5-SPLIT MNIST

We first present our results for class incremental learning of MNIST (5-Split MNIST) in which we learn the digits $0 - 9$ in five tasks with 2 classes at a time in 5 pairs of $0/1$, $2/3$, $4/5$, $6/7$, and $8/9$. Table 2a shows the results for reference baselines in Bayesian and non-Bayesian neural networks including fine-tuning (BBB-FT, ORD-FT), feature extraction (BBB-FE, ORD-FE) and, joint training (BBB-JT, ORD-JT) averaged over 3 runs and standard deviations are given in Table 9 in the appendix. Although the MNIST dataset is an "easy" dataset, we observe throughout all experiments that Bayesian fine-tuning and joint training perform significantly better than their counterparts, ORD-FT and ORD-JT. For Bayesian methods, we compare against VCL and its variations named as VCL with Variational Adam (VCL-Vadam), VCL with Adam and Gaussian natural gradients (VCL-GNG). For non-Bayesian methods, we compare against HAT, IMM, and EWC (EWC can be regarded as Bayesian-inspired). VCL-Vadam (ACC=99.17%) appears to be outperforming VCL (ACC=98.20%) and VCL-GNG (ACC=96.50%) in average accuracy. However, full comparison is not possible because forgetting was not reported for Vadam and GNG. Nevertheless, UCB (ACC=99.63%) is able to surpass all the baselines including VCL-Vadam in average accuracy while in zero forgetting it is on par with HAT (ACC=99.59%). We also report results on incrementally learning MNIST in two tasks (2-Split MNIST) in Table 8 in the appendix, where we compare it

against PackNet, HAT, and LWF where PackNet, HAT, UCB-P, and UCB have zero forgetting while UCB has marginally higher accuracy than all others.

## 5.3 PERMUTED MNIST

Permuted MNIST is a popular variant of the MNIST dataset to evaluate continual learning approaches in which each task is considered as a random permutation of the original MNIST pixels. Following the literature, we learn a sequence of 10 random permutations and report average accuracy at the end. Table 2b shows ACC and BWT of UCB and UCB-P in comparison to state-of-the-art models using a small and a large network with 0.1M and 1.9M parameters, respectively (architecture details are given in Section A.2 of the appendix). The accuracy achieved by UCB (ACC=$91.44 \pm 0.04\%$) using the small network outperforms the ACC reported by Serra et al. (2018) for SI (ACC=$86.0\%$), EWC (ACC=$88.2\%$), while HAT attains a slightly better performance (ACC=$91.6\%$). Comparing the average accuracy reported in VCL-Vadam (ACC=$86.34\%$) and VCL-GNG (ACC=$90.50\%$) as well as obtained results for VCL (ACC=$88.80\%$) shows UCB with BWT=($0.03\% \pm 0.00\%$) is able to outperform other Bayesian approaches in accuracy while forgetting significantly less compared to VCL with BWT=$-7.9\%$. While we do not experiment with memory in this work, not surprisingly adding memory to most approaches will improve their performance significantly as it allows looking into past tasks. E.g. Chen et al. (2019) report ACC=$94.37\%$ for VCL-GNC when adding a memory of size 200.

Next, we compare the results for the larger network (1.9M). While HAT and UCB have zero forgetting, UCB, reaching ACC=$97.42 \pm 0.01\%$, performs better than all baselines including HAT which obtains ACC=$97.34 \pm 0.05\%$ using 1.9M parameters. We also observe again that BBB-FT, despite being not specifically penalized to prevent forgetting, exhibits reasonable negative BWT values, performing better than IMM and LWF baselines. It is close to joint training, BBB-JT, with ACC=$98.1\%$, which can be seen as an upper bound.

## 5.4 ALTERNATING CIFAR10 AND CIFAR100

In this experiment, we randomly alternate between class incremental learning of CIFAR10 and CIFAR100. Both datasets are divided into 5 tasks each with 2 and 20 classes per task, respectively. Table 2c presents ACC and BWT obtained with UCB-P, UCB, and three BBB reference methods compared against various continual learning baselines. Among the baselines presented in Table 2c, PNN and PathNet are the only zero-forgetting-guaranteed approaches. It is interesting to note that in this setup, some baselines (PathNet, LWF, and LFL) do not perform better than the naive accuracy achieved by feature extraction. PathNet suffers from bad pre-assignment of the network's capacity per task which causes poor performance on the initial task from which it never recovers. IMM performs almost similar to fine-tuning in ACC, yet forgets more. PNN, EWC, and HAT are the only baselines that perform better than BBB-FE and BBB-FT. EWC and HAT are both allowed to forget by construction, however, HAT shows zero forgetting behavior. While EWC is outperformed by both of our UCB variants, HAT exhibits $1\%$ better ACC over UCB-P. Despite having a slightly higher forgetting, the overall accuracy of UCB is higher, reaching $79.4\%$. BBB-JT in this experiment achieves a positive BWT which shows that learning the entire sequence improves the performance on earlier tasks.

## 5.5 MULTIPLE DATASETS LEARNING

Finally, we present our results for continual learning of 8 tasks using UCB-P and UCB in Table 2d. Similar to the previous experiments we look at both ACC and BWT obtained for UCB-P, UCB, BBB references (FT, FE, JT) as well as various baselines. Considering the ACC achieved by BBB-FE or BBB-FT ($58.1\%$) as a lower bound we observe again that some baselines are not able to do better than BBB-FT including LFL, PathNet, LWF, IMM, and EWC while PNN and HAT remain the only strong baselines for our UCB-P and UCB approaches. UCB-P again outperforms PNN by $3.6\%$ in ACC. HAT exhibits only $-0.1\%$ BWT, but our UCB achieves $2.4\%$ higher ACC.

## 6 SINGLE HEAD AND GENERALIZED ACCURACY OF UCB

UCB can be used even if the task information is not given at test time. For this purpose, at training time, instead of using a separate fully connected classification head for each task, we use a single

Table 3: Single Head vs. Multi-Head architecture and Generalized vs. Standard Accuracy. Generalized accuracy means that task information is not available at test time. SM, PM, CF, and 8T denote the 5-Split MNIST, Permuted MNIST, Alternating CIFAR10/100, and sequence of 8 tasks, respectively.

| | Generalized ACC | | ACC | | | |
| | Single Head | | Single Head | | Multi Head | |
| Exp | UCB | BBB-FT | UCB | BBB-FT | UCB | BBB-FT |
|---|---|---|---|---|---|---|
| SM | 98.7 | 98.1 | 98.9 | 98.7 | 99.2 | 98.4 |
| PM | 92.5 | 86.1 | 95.1 | 88.3 | 97.7 | 90.0 |
| CF | 71.2 | 65.2 | 74.3 | 67.8 | 79.4 | 68.9 |
| 8T | 76.8 | 47.6 | 79.9 | 53.2 | 84.0 | 43.1 |

head with the total number of outputs for all tasks. For example in the 8-dataset experiment we only use one head with 293 number of output classes, rather than using 8 separate heads, during training and inference time.

Table 3 presents our results for UCB and BBB-FT trained with a single head against having a multi-head architecture, in columns 4-7. Interestingly, we see only a small performance degrade for UCB from training with multi-head to a single head. The ACC reduction is $0.3\%$, $2.6\%$, $5.1\%$, and $4.1\%$ for 2-Split MNIST, Permuted MNIST, Alternating CIFAR10/100, and sequence of 8 tasks experiments, respectively.

We evaluated UCB and BBB-FT with a more challenging metric where the prediction space covers the classes across all the tasks. Hence, confusion of similar class labels across tasks can be measured. Performance for this condition is reported as Generalized ACC in Table 3 in columns 2-3. We observe a small performance reduction in going from ACC to Generalized ACC, suggesting non-significant confusion caused by the presence of more number of classes at test time. The performance degradation from ACC to Generalized ACC is $0.2\%$, $2.6\%$, $3.1\%$, and $3.1\%$ for 2-Split MNIST, Permuted MNIST, Alternating CIFAR10/100, and sequence of 8 tasks, respectively. This shows that UCB can perform competitively in more realistic conditions such as unavailability of task information at test time. We believe the main insight of our approach is that instead of computing additional measurements of importance, which are often task, input or output dependent, we directly use predicted weight uncertainty to find important parameters. We can freeze them using a binary mask, as in UCB-P, or regularize changes conditioned on current uncertainty, as in UCB.

## 7 CONCLUSION

In this work, we propose a continual learning formulation with Bayesian neural networks, called UCB, that uses uncertainty predictions to perform continual learning: important parameters can be either fully preserved through a saved binary mask (UCB-P) or allowed to change conditioned on their uncertainty for learning new tasks (UCB). We demonstrated how the probabilistic uncertainty distributions per weight are helpful to continually learning short and long sequences of benchmark datasets compared against baselines and prior work. We show that UCB performs superior or on par with state-of-the-art models such as HAT (Serra et al., 2018) across all the experiments. Choosing between the two UCB variants depends on the application scenario: While UCB-P enforces no forgetting after the initial pruning stage by saving a small binary mask per task, UCB does not require additional memory and allows for more learning flexibility in the network by allowing small forgetting to occur. UCB can also be used in a single head setting where the right subset of classes belonging to the task is not known during inference leading to a competitive model that can be deployed where it is not possible to distinguish tasks in a continuous stream of the data at test time. UCB can also be deployed in a single head scenario and where tasks information is not available at test time.

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

## A APPENDIX

### A.1 DATASETS

Table 4 shows a summary of the datasets utilized in our work along with their size and number of classes. In all the experiments we resized images to $32 \times 32 \times 3$ if necessary. For datasets with monochromatic images, we replicate the image across all RGB channels.

Table 4: Utilized datasets summary

| Names | #Classes | Train | Test |
|---|---|---|---|
| FaceScrub (Ng & Winkler, 2014) | 100 | 20,600 | 2,289 |
| MNIST (LeCun et al., 1998) | 10 | 60,000 | 10,000 |
| CIFAR100 (Krizhevsky & Hinton, 2009) | 100 | 50,000 | 10,000 |
| NotMNIST (Bulatov, 2011) | 10 | 16,853 | 1,873 |
| SVHN (Netzer et al., 2011) | 10 | 73,257 | 26,032 |
| CIFAR10 (Krizhevsky & Hinton, 2009) | 10 | 39,209 | 12,630 |
| TrafficSigns (Stallkamp et al., 2011) | 43 | 39,209 | 12,630 |
| FashionMNIST (Xiao et al., 2017) | 10 | 60,000 | 10,000 |

### A.2 IMPLEMENTATION DETAILS

In this section we take a closer look at elements of our UCB model on MNIST and evaluate variants of parameter regularization, importance measurement, as well as the effect of the number of samples drawn from the posited posterior.

**Bayes-by-backprop (BBB) Hyperparamters:** Table 5 shows the search space for hyperparamters in the BBB algorithm Blundell et al. (2015) which we used for tuning on the validation set of the first two tasks.

Table 5: Search space for hyperparamters in BBB given by Blundell et al. (2015)

| BBB hyperparamters | $-\log \sigma_1$ | $-\log \sigma_2$ | $\pi$ |
|---|---|---|---|
| Search space | $\{0, 1, 2\}$ | $\{6, 7, 8\}$ | $\{0.25, 0.5, 0.75\}$ |

**Network architecture:** For Split MNIST and Permuted MNIST experiments, we have used a two-layer perceptron which has 1200 units. Because there is more number of parameters in our Bayesian neural network compared to its equivalent regular neural net, we ensured fair comparison by matching the total number of parameters between the two to be 1.9M unless otherwise is stated. For the multiple datasets learning scenario, as well as alternating incremental CIFAR10/100 datasets, we have used a ResNet18 Bayesian neural network with 7.1-11.3M parameters depending on the experiment. However, the majority of the baselines provided in this work are originally developed using some variants of AlexNet structure and altering that, e.g. to ResNet18, resulted in degrading in their reported and experimented performance as shown in Table 6. Therefore, we kept the architecture for baselines as AlexNet and ours as ResNet18 and only matched their number of parameters to ensure having equal capacity across different approaches.

Table 6: Continually learning on CIFAR10/100 using AlexNet and ResNet18 for UCB (our method) and HAT (Serra et al., 2018). BWT and ACC in %. All results are (re)produced by us.

| Method | BWT | ACC |
|---|---|---|
| HAT (AlexNet) | 0.0 | 78.3 |
| HAT (ResNet18) | −9.0 | 56.8 |
| **UCB (AlexNet)** | −0.7 | 79.44 |
| **UCB (ResNet18)** | −0.7 | 79.70 |

**Number of Monte Carlo samples:** UCB is ensured to be robust to random noise using multiple samples drawn from posteriors. Here we explore different number of samples and the effect on final performance for ACC and BWT. We have used $\Omega_\mu = 1/\sigma$ as importance and regularization has been performed on mean values only. Following the result in Table 7 we chose the number of samples to be 10 for all experiments.

Table 7: Number of Monte Carlo samples (N) in 2-Split MNIST

| Method | $N$ | BWT (%) | ACC (%) |
|--------|-----|---------|---------|
| UCB | 1 | 0.00 | 98.0 |
| UCB | 2 | 0.00 | 98.3 |
| UCB | 5 | $-0.15$ | 99.0 |
| UCB | 10 | 0.00 | 99.2 |
| UCB | 15 | $-0.01$ | 98.3 |

## A.3 ADDITIONAL RESULTS

Here we include some additional results such as Table 8 for 2-split MNIST and some complementary results for tables in the main text as follows: 9, 10, and 11 include standard deviation for results shown in Table 2a, 2b, 2c, respectively.

Table 8: Continually learning on 2-Split MNIST. BWT and ACC in %. (*) denotes that methods do not adhere to the continual learning setup: BBB-JT and ORD-JT serve as the upper bound for ACC for BBB/ORD networks, respectively. All results are (re)produced by us.

| Method | BWT | ACC |
|--------|-----|-----|
| PackNet (Mallya & Lazebnik, 2018) | $0.04 \pm 0.01$ | $98.91 \pm 0.03$ |
| LWF (Li & Hoiem, 2016) | $-0.22 \pm 0.04$ | $99.12 \pm 0.03$ |
| HAT (Serra et al., 2018) | $0.01 \pm 0.00$ | $99.02 \pm 0.00$ |
| ORD-FT | $-6.81 \pm 0.03$ | $92.42 \pm 0.02$ |
| ORD-FE | $0.04 \pm 0.04$ | $97.90 \pm 0.04$ |
| BBB-FT | $-0.61 \pm 0.03$ | $98.44 \pm 0.03$ |
| BBB-FE | $0.02 \pm 0.05$ | $98.03 \pm 0.05$ |
| UCB-P (Ours) | $0.03 \pm 0.04$ | $99.02 \pm 0.01$ |
| **UCB (Ours)** | $0.01 \pm 0.00$ | $\mathbf{99.18 \pm 0.01}$ |
| ORD-JT* | $0.02 \pm 0.03$ | $99.13 \pm 0.03$ |
| BBB-JT* | $0.03 \pm 0.02$ | $99.51 \pm 0.02$ |

Table 9: Continually learning on 5-Split MNIST. BWT and ACC in %. (*) denotes that methods do not adhere to the continual learning setup: BBB-JT and ORD-JT serve as the upper bound for ACC for BBB/ORD networks, respectively. All results are (re)produced by us.

| Method | BWT | ACC |
|--------|-----|-----|
| VCL-Vadam (Tseran et al., 2018) | - | $99.17 \pm 0.05$ |
| VCL-GNG (Chen et al., 2019) | - | $96.50 \pm 0.07$ |
| VCL (Nguyen et al., 2018) | $-0.56 \pm 0.03$ | $98.20 \pm 0.03$ |
| IMM (Lee et al., 2017) | $-11.20 \pm 1.57$ | $88.54 \pm 1.56$ |
| EWC (Kirkpatrick et al., 2017) | $-4.20 \pm 1.08$ | $95.78 \pm 1.08$ |
| HAT (Serra et al., 2018) | $0.00 \pm 0.02$ | $99.59 \pm 0.02$ |
| ORD-FT* | $-9.18 \pm 1.12$ | $90.60 \pm 1.12$ |
| ORD-FE* | $0.00 \pm 1.56$ | $98.54 \pm 1.57$ |
| BBB-FT* | $-6.45 \pm 1.99$ | $93.42 \pm 1.98$ |
| BBB-FE* | $0.00 \pm 2.23$ | $98.76 \pm 2.23$ |
| UCB-P (Ours) | $-0.72 \pm 0.04$ | $99.32 \pm 0.04$ |
| **UCB (Ours)** | $0.00 \pm 0.04$ | $\mathbf{99.63 \pm 0.03}$ |
| ORD-JT* | $0.00 \pm 0.02$ | $99.78 \pm 0.02$ |
| BBB-JT* | $0.00 \pm 0.01$ | $99.87 \pm 0.01$ |

Table 10: Continually learning on Permuted MNIST. BWT and ACC in %. (*) denotes that method does not adhere to the continual learning setup: BBB-JT serves as the upper bound for ACC for BBB network. ‡ denotes results reported by (Serra et al., 2018). † denotes the result reported from original work. BWT was not reported in ‡ and †. All others results are (re)produced by us.

| Method | #Params | BWT | ACC |
|---|---|---|---|
| SI (Zenke et al., 2017)[‡] | 0.1M | - | 86.0 |
| EWC (Kirkpatrick et al., 2017)[‡] | 0.1M | - | 88.2 |
| HAT (Serra et al., 2018)[‡] | 0.1M | - | 91.6 |
| VCL-Vadam† | 0.1M | - | 93.34 |
| VCL-GNG† | 0.1M | - | 94.62 |
| VCL | 0.1M | $-7.90 \pm 0.23$ | $88.80 \pm 0.23$ |
| UCB (Ours) | 0.1M | $-0.38 \pm 0.02$ | $91.44 \pm 0.04$ |
| LWF (Li & Hoiem, 2016) | 1.9M | $-31.17 \pm 0.05$ | $65.65 \pm 0.05$ |
| IMM (Lee et al., 2017) | 1.9M | $-7.14 \pm 0.07$ | $90.51 \pm 0.08$ |
| HAT (Serra et al., 2018) | 1.9M | $0.03 \pm 0.05$ | $97.34 \pm 0.05$ |
| BBB-FT | 1.9M | $-0.58 \pm 0.05$ | $90.01 \pm 0.05$ |
| BBB-FE | 1.9M | $0.02 \pm 0.03$ | $93.54 \pm 0.04$ |
| UCB-P (Ours) | 1.9M | $-0.95 \pm 0.06$ | $97.24 \pm 0.06$ |
| **UCB (Ours)** | 1.9M | $0.03 \pm 0.00$ | $\mathbf{97.42 \pm 0.01}$ |
| BBB-JT[*] | 1.9M | $0.00 \pm 0.00$ | $98.12 \pm 0.01$ |

Table 11: Continually learning on CIFAR10/100. BWT and ACC in %. (*) denotes that method does not adhere to the continual learning setup: BBB-JT serves as the upper bound for ACC for BBB network. All results are (re)produced by us.

| Method | BWT | ACC |
|---|---|---|
| PathNet (Fernando et al., 2017) | $0.00 \pm 0.00$ | $28.94 \pm 0.03$ |
| LWF (Li & Hoiem, 2016) | $-37.9 \pm 0.32$ | $42.93 \pm 0.30$ |
| LFL (Jung et al., 2016) | $-24.22 \pm 0.21$ | $47.67 \pm 0.22$ |
| IMM (Lee et al., 2017) | $-12.23 \pm 0.06$ | $69.37 \pm 0.06$ |
| PNN (Rusu et al., 2016) | $0.00 \pm 0.00$ | $70.73 \pm 0.08$ |
| EWC (Kirkpatrick et al., 2017) | $-1.53 \pm 0.07$ | $72.46 \pm 0.06$ |
| HAT (Serra et al., 2018) | $0.04 \pm 0.06$ | $78.32 \pm 0.06$ |
| BBB-FE | $0.04 \pm 0.02$ | $51.04 \pm 0.03$ |
| BBB-FT | $-7.43 \pm 0.07$ | $68.89 \pm 0.07$ |
| UCB-P (Ours) | $-1.89 \pm 0.03$ | $77.32 \pm 0.03$ |
| **UCB (Ours)** | $-0.72 \pm 0.02$ | $\mathbf{79.44 \pm 0.02}$ |
| BBB-JT[*] | $1.52 \pm 0.04$ | $83.93 \pm 0.04$ |

