# OpenReview forum: "Uncertainty-guided Continual Learning with Bayesian Neural Networks"
_ICLR.cc/2020/Conference — Accept (Poster)_

### Official Review · AnonReviewer2 · 2019-10-23
**Official Blind Review #2**

**Rating:** 6

**Review:**

**** Post Rebuttal ****

I have read the author's response and other reviewers' comments. In light of comments by other reviewers, I am increasing the score. The paper reports decent empirical results in some challenging settings which might be useful to the continual learning community.

**** End ****

The paper presents a simple yet effective way to avoid catastrophic forgetting in a continual learning setting. The proposed approach is referred to as UCB - "Uncertainty Guided Bayesian Neural Networks". The main idea of the approach is to weight the learning rate of each parameter in the neural network by the standard deviation of its posterior distribution. This leads to regularizing parameters that are "important" to tasks seen earlier and thus avoiding forgetting.  Results indicate an improvement over other baselines. However, I do not see any analysis of the method that explains this improvement. I do not recommend acceptance.

Cons:

- My main concern with the paper is that it fails to justify the superiority of the method over other baselines. The numbers reported in the paper do seem good, but I don't see an explanation of why this is the case. What are the drawbacks of EWC, VCL or HAT that the proposed method solves? Why using uncertainty to define importance works better than using online VI in VCL or fisher information in EWC? There is no discussion in the paper about that. Without such a discussion it seems that the model was run a number of times and the best score was reported out of all those runs (especially because the improvement is only marginal).

- I am not sure why weighting the learning rate would be a good idea? Having high uncertainty may increase the learning rate arbitrarily. Is there a constraint on the standard deviation? Does having a very high weight for learning rate not cause instability during optimization? I think the method would be very sensitive to the initialization of the standard deviation.

Overall I think the idea of using uncertainties for continual learning is interesting. But from where it stands, I am not fully convinced that this method should do better than existing approaches.

**Experience Assessment:**

I have read many papers in this area.

**Review Assessment: Checking Correctness Of Derivations And Theory:**

N/A

**Review Assessment: Checking Correctness Of Experiments:**

I assessed the sensibility of the experiments.

**Review Assessment: Thoroughness In Paper Reading:**

I made a quick assessment of this paper.

---

> ### Author Response · Authors · 2019-11-15
> **Response to R2 -- Part 1**
>
> We thank the reviewer for their comments and are happy to hear that they find our idea of using uncertainty interesting.In the following we address the individual comments:
> =================================================
> 1. The reviewer is concerned that our paper “fails to justify the superiority of the method over other baselines” and misses and "explanation of why this is the case”.
>
> While it is not fully clear to us how “superiority” is defined we believe our approach has the following properties, which make it valuable and interesting:
> Our approach is novel (R#1)
> Our approach is “simple but effective” (R#3)
> Our approach makes sense, as it follows Bayesian principles of uncertainty, which means the uncertainty is inherent to the model which we use to define  importance. See Section 1, 4th paragraph “Bayesian approaches to …” and Figure 1, for more on the motivation.
> R1 states this “work is highly significant” and is supported with an experimental evaluation “with a very large number of baselines”
> Our approach is *different* from prior work (as discussed extensively in Section 2); some aspects proposed in prior work are orthogonal to our work, such as usage of episodic memory or model growth; others are just different, e.g. many regularization based methods use an additional “external” importance parameter for each network parameter, while we exploit the “additional” \rho parameters inherently to Bayesian Neural Network without the need of an “external” importance parameter.
> [we don’t claim our approach is “superior” to all other prior work w.r.t. methodology; however, it is simple and well motivated and experimentally we find that our performance is very competitive (on par or better than prior work) on a broad set of experiments].
>
> Overall, given these aspects, we strongly believe our approach will be appreciated by the community.
>
> =================================================
> “What are the drawbacks of EWC, VCL or HAT that the proposed method solves?”
>
> Our UCB is based on Bayesian neural networks and exploits their inherent uncertainty modeling to change the learning rate per parameter.
>
> HAT is regularization based but does not use a Bayesian Neural Network.
> EWC is a Bayesian-inspired method but does not rely on Bayesian Neural Networks, i.e. it does not exploit the inherent uncertainty modeling Bayesian Neural Networks.
> VCL uses Bayesian inference, in contrast UCB is based on Bayesian neural networks to use their predictive uncertainty to perform continual learning.
>
> Section 2, gives a detailed discussion to prior work, and we experimentally support the strength of our method in the paper.
> Additionally w.r.t. Bayesian continual learning methods: None of them have been applied on CNNs so we are the only work that have extended it to real world images in a long sequence of tasks. See also our challenging 8 task experiment.
>
> =================================================
> “Why using uncertainty to define importance works better than using online VI in VCL or fisher information in EWC?”
>
> We believe this is because the uncertainty in Bayesian Neural Networks gives a good estimate for parameter importance used in continual learning.
> This is clearly different than VI in VCL or the fisher information in EWC and while we do not have a mathematical proof for being better (which might also be difficult), we find our experiments support our hypothesis that using uncertainty in Bayesian Neural Networks is a good idea.
>
> =================================================
> “[...] it seems that the model was run a number of times and the best score was reported out of all those runs (especially because the improvement is only marginal).”
>
> We like to highlight our very restrictive experimental setup we employ (in contrast to most prior work in continual learning). We only rely on the first two tasks and their validations set to tune hyperparameters, similar to the setup in (Chaudhry et al., 2019). (see “Hyperparameter tuning” in Section 5.1).
> We do not report “the best score“ but the average over multiple runs in the main paper, and, in the appendix (section A.3) we also show standard deviation (Tables 8, 9, 10, 11).

---

> > ### Author Response · Authors · 2019-11-15
> > **Response to R2 -- Part 2**
> >
> > 2.  “I am not sure why weighting the learning rate would be a good idea?”
> >
> > To us, it seems a very natural and obvious choice. Decreasing the learning rate for important parameters decreases changing them which results in not forgetting previous tasks.
> >
> > =================================================
> > “Having high uncertainty may increase the learning rate arbitrarily. Is there a constraint on the standard deviation? Does having a very high weight for learning rate not cause instability during optimization? I think the method would be very sensitive to the initialization of the standard deviation. “
> >
> > We follow the initialization strategy used in the original Bayes-by-Backprop (BBB) framework (Blundell et al., 2015). The right initialization is important and we treat it as a hyperparameter which we find, as all other hyperparameters (see above) with the validation sets of the first two tasks. We added a paragraph Bayes-by-backprop (BBB) Hyperparamters: in Section A.2 in appendix  in the revised pdf detailing this. When initialized correctly, \rho is not exploding in our experiments with BBB.
> >
> > ====

---

### Official Review · AnonReviewer3 · 2019-10-23
**Official Blind Review #3**

**Rating:** 6

**Review:**

** post rebuttal start **

After reading reviews and authors' response, I decided not to change my score.
I am happy with the author's response addressing my concerns (mainly about the fairness on the size of the model), so I recommend its acceptance. I believe it is a good addition to the community of continual learning.

** post rebuttal end **


- Summary:
This paper proposes to use a way to improve continual learning performance by taking "Bayes-by-backprop" method. They claim that the uncertainty can naturally be measured by estimating (log of) the standard deviation, and it is indeed useful to judge the importance of each learnable parameter. Experimental results on several benchmarks show that their method outperforms few state-of-the-art methods.


- Decision and supporting arguments:
Weak accept.

1. The proposed method is simple but effective. However, It is still questionable whether \sigma is the best measure of the weight importance. An ablation study with different choices of the importance measure (maybe \mu can also be incorporated as well as \sigma?) would be good to see.

2. Survey and comparison with memory-based methods are limited. Though memory-based methods require some memory to keep the experience, the proposed method also requires additional memory for \sigma; it essentially doubles the model capacity, assuming that \sigma is solely for measuring the weight importance. In particular, when it comes to large-scale models, memory for storing some important experiences would be small compared to the memory to store the model.
Here are some papers about recently proposed memory-based methods, which are not cited:

Castro et al. End-to-End Incremental Learning. In ECCV, 2018.
Wu et al. Large Scale Incremental Learning. In CVPR, 2019.
Lee et al. Overcoming Catastrophic Forgetting with Unlabeled Data in the Wild. In ICCV, 2019.

3. Comparison should include the model capacity as in Table 1(b). Again, compared to the conventional non-Bayesian model, half of the model capacity is used for computing \sigma (uncertainty), I wonder it causes a performance drop when the model capacity is the same over all compared methods. If they used the same model architecture and just doubled the number of learnable parameters for \sigma, then it is obviously unfair.


- Comments:
1. Pruning is not beneficial in terms of the performance. I hope to see some quantitative benefits obtained by introducing pruning. In Table 1(b), why doesn't pruning reduce the number of parameters?


**Experience Assessment:**

I have published one or two papers in this area.

**Review Assessment: Checking Correctness Of Derivations And Theory:**

I assessed the sensibility of the derivations and theory.

**Review Assessment: Checking Correctness Of Experiments:**

I assessed the sensibility of the experiments.

**Review Assessment: Thoroughness In Paper Reading:**

I read the paper at least twice and used my best judgement in assessing the paper.

---

> ### Author Response · Authors · 2019-11-15
> **Response to R3**
>
> We thank the reviewer for his/her comments about our work. We reply to the comments in chronological order:
>
> 1. We have already provided this ablation in Table 5 in the appendix in which we considered other variants for the weight importance: specifically, we look into regularizing \mu and \rho or both and explore if 1/\sigma or |\mu|/\sigma is better for importance measurement. We find that highest accuracy and BWT is achieved by 1/\mu for  UCB and |\mu|/\sigma for UCB-P, but other variants don’t decrease the performance dramatically. We have moved it to the main text as Table 1 on page 7 in the revised version.
>
> 2. We agree with the reviewer that the memory of the entire model should be taken into account. And we do that, as we detail in section A2. We make sure our UCB matches to the baselines w.r.t. the *total* number of parameters (the sum of \mu and \rho for UCB). In table 1b we also list the *total* number of parameters (For UCB-P the memory for the mask is not included, see also 4. below).
> We agree it might be a good option to compare methods by the total memory usage, when comparing regularization with episodic memory based models.
> While there are reasons for and against episodic memory storage (e.g. potential privacy concerns, even when just storing representations), this is not the focus of this work and orthogonal to this work. In fact we believe our approach would benefit from episodic memory, especially in the challenging setting of single head and generalized accuracy (section 6). We leave the exploration of combining UCB with episodic memory to future work.
> We also cited the mentioned references in the updated draft in the related work, section 2 under “memory-based methods” subsection.
>
> 3. We agree with the reviewer and we have, compared to baselines, used *already* only half of the number of weights for UCB, as each weight consist of 2 parameters (see also reply to 2.).  In section A.2 we have detailed this aspect; we ensured a fair comparison by matching the *total* number of learnable parameters.
>
> 4. (Question 1 in comments): Table 1b: For UCB-P the #params match the total number of parameters initially, but also after training for the last task. The reason for this is that we do not prune the network anymore after training the last task (we would do that when the next task arrives). However, we like to note that UCB-P, by using a “hard” binary mask for each task will use up more and more parameters for each new task it sees which it cannot free by pruning. So when arriving at the last task only relatively few parameters are remaining, the ability to further prune is thus limited.
> We detail our pruning procedure in Section 5.1,  in the paragraph, “Pruning procedure and mask size” where we explain what percentage of network is pruned at each time. We agree with the reviewer that pruning (UCB-P) is not efficient and in our experiments it yields lower performance compared to our soft regularization version (UCB) which we introduced as our main method. However, in case it is desired to recover the “exact same performance” post-pruning, one might consider using it because soft regularization methods are not zero-forgetting guaranteed. We also want to mention that pruning techniques are not really “zero-forgetting” because the accuracy drop during pruning can be considered as forgetting as we do in this paper.

---

### Official Review · AnonReviewer1 · 2019-10-25
**Official Blind Review #1**

**Rating:** 8

**Review:**

The authors propose a novel method for continual learning with neural networks based on a Bayesian approach. The idea consists in working with Bayesian neural networks, using the Bayes by back-prop approach in which a factorized Gaussian variational distribution is used to approximate the true posterior. To address the continual learning setting, the authors propose to multiply the learning rate of the mean parameters in the posterior approximation by the corresponding standard deviation parameter in the posterior approximation, while the learning rate for the variance parameters in the posterior approximation is not changed. The authors also consider a version of his method which freezes the mean and variance variational parameters when the signal to noise ratio is high. The proposed method is evaluated in exhaustive experiments, showing state-of-the-art results.

Clarity:

The paper is clearly written and easy to read. The method proposed is well described and it would be easy to reproduce.

Quality:

The proposed method is well justified and the experiments performed clearly illustrate the gains with respect to previous methods.

Novelty:

The proposed method is novel up to my knowledge. The methodological contributions do not seem very sophisticated, but the experiments show that the proposed method, despite being very simple, works very well in practice.

Significance:

The experiments show that the proposed method achieves state of the art results when compared with a very large number of baselines. This indicates that the proposed method will be relevant to the community. In my opinion, this work is highly significant.

**Experience Assessment:**

I have read many papers in this area.

**Review Assessment: Checking Correctness Of Derivations And Theory:**

I assessed the sensibility of the derivations and theory.

**Review Assessment: Checking Correctness Of Experiments:**

I assessed the sensibility of the experiments.

**Review Assessment: Thoroughness In Paper Reading:**

I read the paper at least twice and used my best judgement in assessing the paper.

---

> ### Author Response · Authors · 2019-11-15
> **Response to R1**
>
> We thank the reviewer for their positive feedback, and are happy that they appreciate the clarity, quality, novelty, and high significance of our work.

---

### Author Response · Authors · 2019-11-15
**See individual comments and revised pdf**

We thank all reviewers for their feedback and we replied to individual reviews directly. We also revised the pdf addressing concerns as discussed in the individual author responses.

---

### Decision · Program_Chairs · 2019-12-19

**Decision:**

Accept (Poster)

**Comment:**

While prior work has shown the potential of using uncertainty to tackle catastrophic forgetting (e.g. by appropriate updates to the posterior), this paper goes further and proposes a strategy to adapt the learning rate based on the uncertainty. This is a very reasonable idea since, in practice, learning rate control is one of the simplest and most understood techniques to fight catastrophic forgetting.
The overall approach ends up being a well-motivated strategy for controlling the learning rate of the parameters according to a notion of their "importance". Of course now the question is if this work uses a good proxy for "importance" so further ablation studies would help, but the current results already show a clear benefit.